# Lobeglitazone inhibits LPS-induced NLRP3 inflammasome activation and inflammation in the liver

Hye-Young Seo[1], So-Hee Lee[1], Ji Yeon Park[1], Eugene Han[1], Sol Han[2], Jae Seok Hwang[1], Mi Kyung Kim[1]*, Byoung Kuk Jang[1]*

**1** Department of Internal Medicine, School of Medicine, Institute for Medical Science, Keimyung University, Daegu, Korea, **2** Department of Physiology, University of Washington, Seattle, WA, United States of America

* mdkmk@dsmc.or.kr (MKK); jangha106@dsmc.or.kr (BKJ)

**Data Availability Statement:** All relevant data are within the paper and its Supporting Information files.

**Funding:** This research was supported by the Basic Science Research Program through the National

## Abstract

Liver inflammation is a common feature of chronic liver disease and is often associated with increased exposure of the liver to lipopolysaccharide (LPS). Kupffer cells (KCs) are macrophages in the liver and produce various cytokines. Activation of KCs through the NLRP3 inflammasome pathway leads to release of proinflammatory cytokines and induces hepatocyte injury and hepatic stellate cell (HSC) activation. Lobeglitazone is a peroxisome proliferator-activated receptor gamma ligand and a type of thiazolidinedione that elicits anti-inflammatory effects. However, there is no clear evidence that it has direct anti-inflammatory effects in the liver. This study showed that lobeglitazone reduces LPS-induced NLPR3 inflammasome activation and production of proinflammatory cytokines in primary KCs and hepatocytes. Cytokines secreted by activated KCs increased hepatocyte inflammation and HSC activation, and lobeglitazone inhibited these responses. In addition, lobeglitazone suppressed liver fibrosis by inhibiting LPS-induced transforming growth factor (TGF)-β secretion and TGF-β-induced CTGF expression. The inhibitory effect of lobeglitazone on inflammasome activation was associated with suppression of liver fibrosis. These results suggest that lobeglitazone may be a treatment option for inflammation and fibrosis in the liver.

## Introduction

Liver inflammation is a common feature of chronic liver disease and is often associated with increased exposure of the liver to lipopolysaccharide (LPS) [1]. An increased level of LPS in the liver causes hepatocyte damage and stimulates liver macrophages [2]. Kupffer cells (KCs) are macrophages in the liver and an essential part of the innate immune system [3]. Alterations in the functional activity of KCs are associated with a variety of liver diseases (e.g., nonalcoholic fatty liver disease and nonalcoholic steatohepatitis) [3–5].

KCs are involved in production of various cytokines such as interleukin 1 (IL1), interleukin 6 (IL6), and tumor necrosis factor (TNF) and regulate inducible nitric oxide synthase (iNOS)

Research Foundation of Korea (NRF) funded by the Ministry of Education (NRF-2021R1I1A3046593, NRF- 2021R1I1A3059150), and was supported by a NRF grant funded by the Korea government (MIST) (NRF-2022R1A2C1006416). (BK Jang, MK Kim, HY Seo).

**Competing interests:** The authors have declared that no competing interests exist.

levels by releasing reactive oxygen species [6, 7]. Cytokines secreted by activated KCs can directly affect the activation of hepatic stellate cells (HSCs) [8]. These factors can contribute to proliferation and activation of HSCs and extracellular matrix (ECM) synthesis and lead to liver fibrosis [9]. In addition, several cytokines secreted by HSCs induce cell-cell interactions by recruiting KCs and subsequently stimulate KCs to release more inflammatory mediators [10, 11].

The NLRP3 inflammasome is a multiprotein complex containing the adapter protein apoptosis-associated speckle-like protein (ASC) and the serine protease caspase 1 [12–14]. Activation of the inflammasome in macrophages triggers an inflammatory response and amplifies hepatocyte damage through a common pathway of the NLRP3 inflammasome and IL1β, contributing to various liver diseases [15, 16]. Therefore, the inflammasome has been increasingly implicated in severe liver inflammation, fibrosis, and cell death [11, 17].

Lobeglitazone, a type of thiazolidinedione (TZD), is a peroxisome proliferator-activated receptor gamma (PPARγ) ligand used to regulate glucose and lipid metabolism in type 2 diabetes [18]. It reportedly attenuates hepatic steatosis in obese mice by improving insulin resistance and inhibiting lipogenesis-induced liver injury [19, 20]. Lobeglitazone also reportedly has an anti-inflammatory effect [21, 22]. However, reports of its anti-inflammatory effect in the liver are limited, and there is no clear evidence that lobeglitazone ameliorates inflammation through direct effects on the liver.

In this study, we aimed to evaluate the effect of lobeglitazone on NLRP3 inflammasome activation and to explore potential underlying mechanisms.

## Materials and methods

### Reagents and antibodies

Lobeglitazone was kindly supplied by Chong Kun Dang Pharmaceutical Corporation (Seoul, Republic of Korea). LPS (*Escherichia coli* 055, B5) was purchased from Sigma-Aldrich (St. Louis, MO, USA). Recombinant human TGF-β (5 ng/mL) was purchased from R&D Systems (Minneapolis, MN, USA). Anti-CTGF (SC365970) and anti-ASC (SC514414) antibodies were purchased from Santa Cruz Biotechnology (Dallas, TX, USA). An anti-IL1β antibody (ab9722) was purchased from Abcam (Cambridge, UK). An anticollagen antibody (PA529569) was purchased from Thermo Fisher Scientific (Waltham, MA, USA). Anti-NLRP3 (CS1510), anticleaved caspase 1 (CS89332), anti-GAPDH (CS2118), antitubulin (CS2146), anti-p-STAT3 (CS9138), and anti-STAT3 (CS4904) antibodies, and antirabbit (7074P2) and antimouse (7076P2) secondary antibodies, were purchased from Cell Signaling Technology (Beverly, MA, USA).

### Isolation of primary KCs and hepatocytes

All experiments were approved by the institutional animal care and use committee of Keimyung University (KM-2020-26R). All animal procedures were performed in strict accordance with institutional guidelines for animal research. All surgeries were performed under sodium pentobarbital anesthesia, and every effort was made to minimize pain. Mouse KCs and hepatocytes were obtained by perfusion of EGTA solution and collagenase solution (collagenase type I; Worthington Biochemical Corp., Lakewood, NJ, USA) through the portal vein of C57BL/6 mice. The liver was then shaken at 37˚C for 20 min, filtered through a 70 μm nylon mesh, and centrifuged at 500 rpm for 5 min to generate a pellet containing hepatocytes and a supernatant containing KCs. Hepatocytes were resuspended in Williams' medium E (Sigma-Aldrich), cultured in type I collagen-coated dishes (IWAKI Scitech Kiv, Tokyo, Japan) for 1–2 h, and then cultured with medium 199 (Sigma-Aldrich). The hepatocyte pellet and the

separated supernatant were centrifuged at 1600 rpm for 10 min, and then the pellet was sub-jected to OptiPrep (Sigma-Aldrich) density-gradient centrifugation to separate KC. The iso-lated KC was plated on RPMI 1640 medium (Gibco-BRL, Grand Island, NY, USA) containing 10% fetal bovine serum and incubated for 30 min, then the media was replaced to obtain puri-fied KC. KCs and hepatocytes were pretreated with compounds in 0.5% FBS with or without LPS (1 μg/mL) for 2 h, followed by lobeglitazone (10 μM) for 24 h.

## Isolation of primary HSCs

HSCs were isolated from C57BL/6 mice. EGTA, pronase (Roche Diagnostics, Indianapolis, IN, USA), and collagenase (Roche Diagnostics) buffers were sequentially perfused through the infe-rior vena cava. The liver was shaken in an incubator at 37°C for 20 min, filtered through a 70 μm nylon mesh, and centrifuged at 1625 rpm for 10 min. The pellet containing isolated hepa-tocytes was mixed with OptiPrep (Sigma-Aldrich) and gently overlaid with a gradient. The sam-ple was centrifuged at 3000 rpm for 17 min at 4°C without braking. HSCs present in the thin white layer at the interface between OptiPrep and Hanks' Balanced Salt Solution (HBSS) were harvested, washed with HBSS, and plated in DMEM (Gibco-BRL) containing 10% FBS.

## Quantitative real-time PCR

Total RNA was isolated from cells using TRIzol reagent (Invitrogen, Waltham, MA, USA), and reverse transcription was performed using a Maxima First Strand cDNA Synthesis Kit (Thermo Fisher Scientific). cDNA was diluted and reacted using SYBR Green PCR Master Mix (Roche Diagnostics) and a CFX Connect Real-Time PCR System (Bio-Rad, Richmond, CA, USA). Primer sequences were as follows: NLRP3 (forward) 5′–ATTACCCGCCCGAGAAA GG–3′ and (reverse) 5′–CATGAGTGTGGCTAGATCCAAG–3′, ASC (forward) 5′–TGCAACTG CGAGAAGGCTAT–3′and (reverse) 5′–GTGAGCTCCAAGCCATACGA–3′, caspase 1 (forward) 5′–CGTACACGTCTTGCCCTCAT–3′and (reverse) 5′–AACTTGAGCTCCAACCCTCG–3′, iNOS (forward) 5′–CATGCTACTGGAGGTGGGTG–3′and (reverse) 5′–CATTGATCTCCGTG ACAGCC–3′, IL1α (forward) 5′–CAACGTCAAGCAACGGGAAG–3′and (reverse) 5′–AAGGT GCTGATCTGGCTTGG–3′, IL1β (forward) 5′–CTTTCCCGTGGACCTTCCAG–3′and (reverse) 5′–AATGGGAACGTCACACACCA–3′, IL6 (forward) 5′–TTGCCTTCTTGGGACTGATG–3′and (reverse) 5′–CTCATTTCCACGATTTCCCA–3′, TNFα (forward) 5′–ACCGTCAGCCGATTTGC TAT–3′and (reverse) 5′–CCGGACTCCGCAAAGTCTAA–3′, TGF-β (forward) 5′–AAATCAA CGGGATCAGCCCC–3′ and (reverse) 5′–GGATCCACTTCCAACCCAGG–3′, and GAPDH (forward) 5′–ACGACCCCTTCATTGACCTC–3′and (reverse) 5′–ATGATGACCCTTTTGGCT CC–3′.

## Measurement of cytokine levels

KCs and hepatocytes were seeded, pretreated with LPS for 2 h, and then treated with lobeglita-zone for 24 h. The cultures were centrifuged, and cell pellets or supernatants (conditioned media, CM) were stored at –80°C. Levels of iNOS (Abcam) and IL1α, IL1β, IL6, TNF, and TGF-β (R&D Systems, Abingdon, UK) were measured using enzyme-linked immunosorbent assay (ELISA) kits, following the manufacturers' instructions.

## Western blotting

Cells were lysed with RIPA lysis buffer (Thermo Fisher Scientific) containing a cocktail of pro-tease/phosphatase inhibitors (genDEPOT, Katy, TX, USA). Proteins were separated by SDS-PAGE and transferred to polyvinylidene fluoride membranes (Millipore, Billerica, MA,

USA). The membranes were blocked with 5% skim milk prepared in Tris-buffered saline containing 0.1% Tween 20) and incubated with primary antibodies followed by an appropriate horseradish peroxidase-conjugated secondary antibody. Protein expression was detected using the Fusion Solo ChemiDoc system (VILBER LOURMAT, Germany). Protein band intensities were measured using ImageJ software version 1.52a (NIH, Bethesda, MD, USA).

### siRNA targeting PPARγ (siPPARγ)

Predesigned siPPARγ was purchased from Santa Cruz Biotechnology. siRNA was transfected using Lipofectamine RNAiMAX reagent (Invitrogen, Carlsbad, CA, USA).

### Statistical analysis

Experimental results were statistically analyzed by a one-way analysis of variance with the Bonferroni correction or the two-tailed student's t test (GraphPad, Prism version 9.5.1). $P < 0.05$ and $P < 0.01$ were considered statistically significant. Data are presented as mean ± standard error of the mean (SEM). All experiments were performed at least three times.

## Results

### Lobeglitazone inhibits secretion of IL1β and other cytokines by LPS-stimulated KCs

To determine whether lobeglitazone inhibits production of proinflammatory cytokines by LPS-stimulated primary KCs, we measured the mRNA levels and secretion of these cytokines. LPS increased mRNA levels and secretion of iNOS, IL6, and TNFα, but these effects were attenuated by lobeglitazone (Fig 1A and 1B). We next determined whether lobeglitazone affects expression and secretion of the inflammasome-specific cytokine IL1β. As expected, mRNA expression and secretion of IL1β were increased in LPS-stimulated KCs, but these effects were attenuated by lobeglitazone (Fig 1A and 1B).

### Lobeglitazone inhibits NLRP3 inflammasome activation in LPS-stimulated KCs

Next, to verify whether the increased secretion of IL1β upon LPS stimulation contributes to NLRP3 inflammasome activation in KCs, we investigated the effect of lobeglitazone on the NLRP3 inflammasome pathway. In KCs, LPS increased mRNA and protein expression of NLRP3, but these effects were significantly attenuated by lobeglitazone (Fig 2A and 2B). Upregulation and activation of NLRP3 promotes activation of caspase 1 and processing and secretion of IL1β. Lobeglitazone significantly attenuated LPS-induced activation of caspase 1 and upregulation of mature IL1β (Fig 2A and 2B). These results indicate that lobeglitazone inhibits activation of the NLRP3 inflammasome.

### Lobeglitazone inhibits inflammation induced by KC-secreted cytokines in hepatocytes

Activated KCs secrete cytokines to increase hepatocyte inflammation [23]. Therefore, we investigated whether lobeglitazone inhibits inflammation induced by KC-secreted cytokines in hepatocytes. mRNA levels of IL1β and the proinflammatory cytokines iNOS, IL1α, IL6, and TNFα were increased in hepatocytes cultured in CM of LPS-treated KCs, but these effects were significantly attenuated by lobeglitazone (Fig 3). Therefore, cytokines secreted by KCs

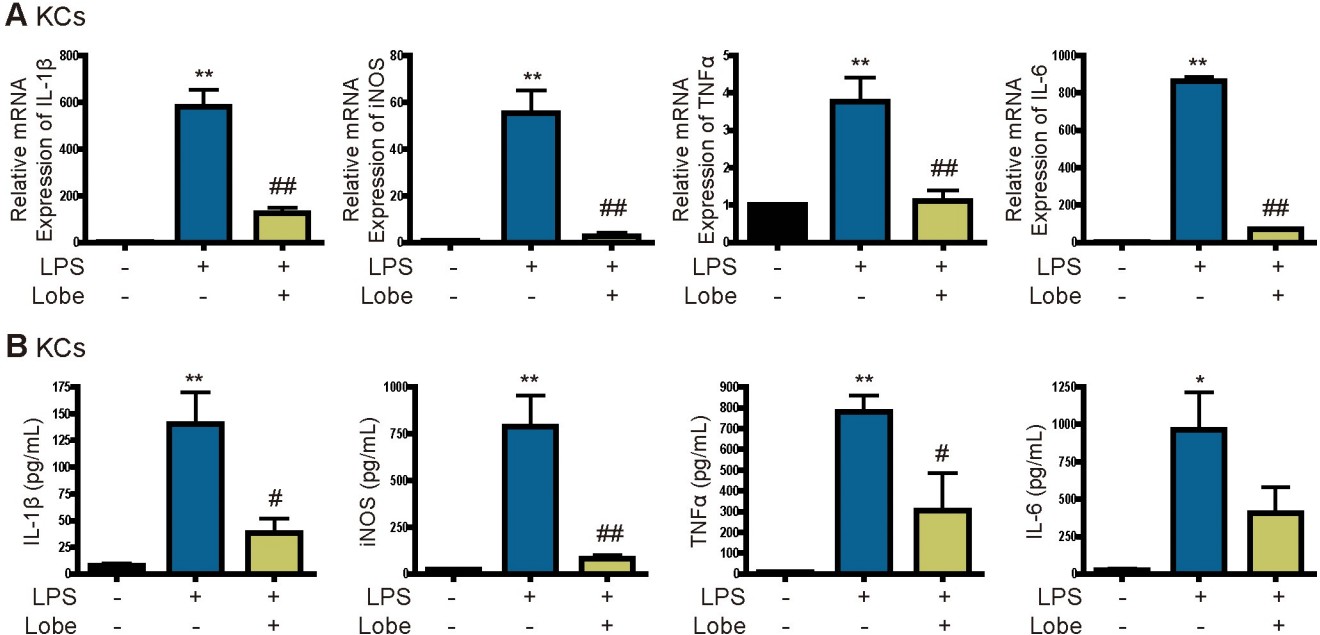

**Fig 1. Lobeglitazone inhibits LPS-induced expression of proinflammatory cytokines in KCs.** Primary KCs were pretreated with LPS for 2 h and then treated with lobeglitazone for 24 h. (A) Representative real-time RT-PCR analysis of the expression levels of IL1β, iNOS, IL6, and TNFα. Data are mean ± SEM. **P < 0.01 versus the control, ##P < 0.01 versus the LPS-treated group. (B) After 24 h, the media were collected and analyzed by ELISAs. Data are mean ± SEM of three independent measurements. *P < 0.05, **P < 0.01 versus the control; #P < 0.05, ##P < 0.01 versus the LPS-treated group.

upregulate the inflammasome-specific cytokine IL1β and proinflammatory activity, but lobeglitazone attenuates these effects.

## Lobeglitazone inhibits NLRP3 inflammasome activation in LPS-stimulated hepatocytes

Next, we examined the effects of lobeglitazone on NLRP3 inflammasome activation in hepatocytes. LPS increased mRNA expression of NLRP3, ASC, and caspase 1, but these effects were significantly attenuated by lobeglitazone (Fig 4A). In addition, the effect of lobeglitazone on secretion of the inflammasome-specific cytokine IL1β was examined. Lobeglitazone decreased LPS-induced secretion of IL1β and expression of NLRP3 and mature IL1β (Fig 4B and 4C). Therefore, lobeglitazone inhibits NLRP3 inflammasome activation in hepatocytes as well as in KCs.

## Lobeglitazone inhibits inflammation induced by KC-secreted cytokines in HSCs

Several cytokines secreted by KCs directly affect HSC activation, and these cells are therefore central regulators in liver fibrosis [8]. We next investigated whether lobeglitazone inhibits inflammation induced by KC-secreted cytokines in primary HSCs. The mRNA levels of NLRP3, ASC, and caspase 1 were higher in HSCs incubated in CM of LPS-treated KCs than in controls, but these effects were significantly attenuated by lobeglitazone (Fig 5A). In addition, the mRNA levels of alpha smooth muscle actin (αSMA) and collagen were higher in HSCs incubated in CM of LPS-treated KCs than in controls, but these effects were significantly attenuated by lobeglitazone (Fig 5B). Next, we investigated whether lobeglitazone directly affects HSC activation. Protein expression of NLRP3, ASC, and cleaved caspase 1 was increased in

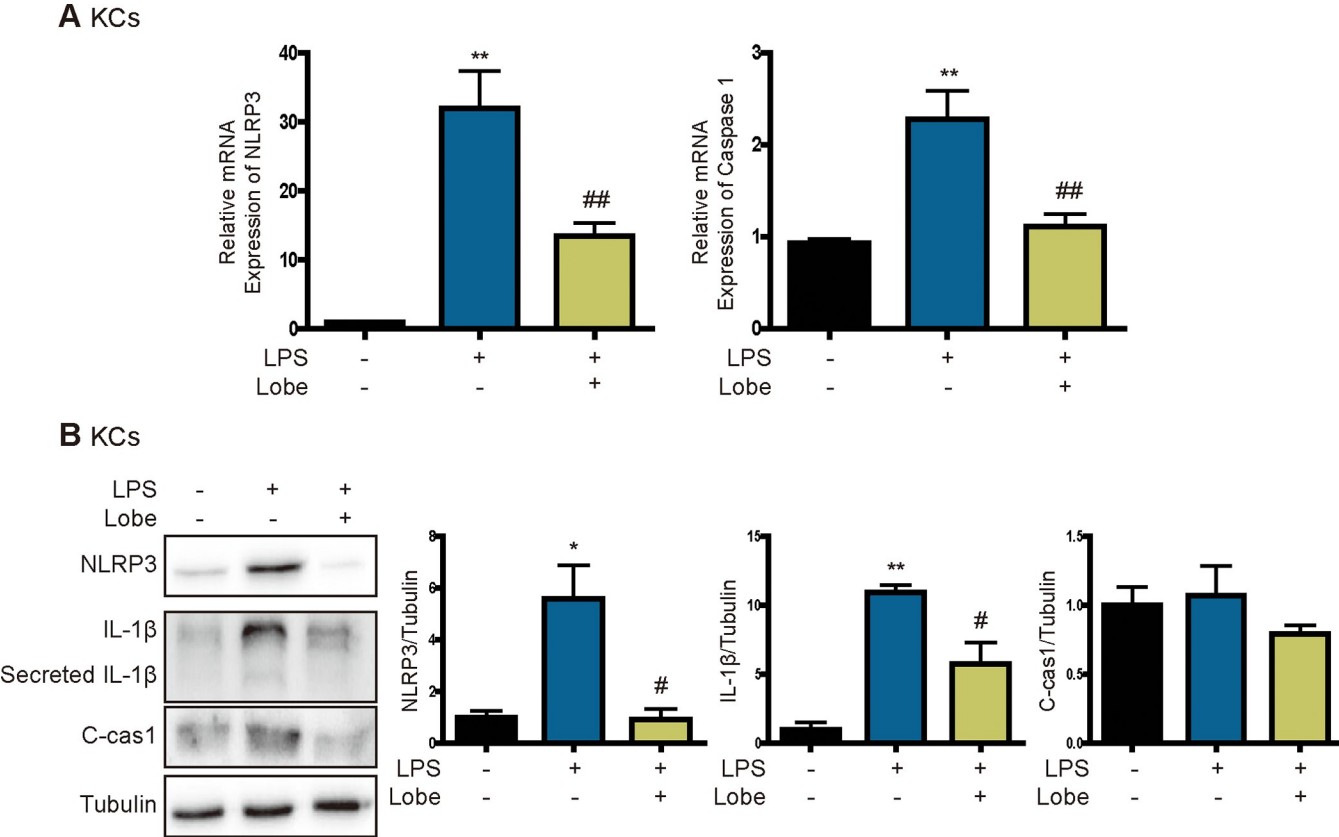

**Fig 2. Lobeglitazone inhibits the LPS-induced NLRP3 inflammasome pathway in KCs.** Primary KCs were pretreated with LPS for 2 h and then treated with lobeglitazone for 24 h. (A) Representative real-time RT-PCR analysis of the expression levels of NLRP3 and caspase 1. Data are mean ± SEM. **$P < 0.01$ versus the control, ##$P < 0.01$ versus the LPS-treated group. (B) Western blot analysis of the effects of lobeglitazone on LPS-induced NLRP3, IL1β, and cleaved caspase 1 expression. Data in the bar graphs are mean ± SEM. *$P < 0.05$, **$P < 0.01$ versus the control; #$P < 0.05$ versus the LPS-treated group.

activated HSCs and reduced by lobeglitazone. In addition, lobeglitazone decreased protein expression of αSMA and collagen (Fig 5C).

## Lobeglitazone inhibits TGF-β, a profibrotic cytokine secreted by KCs and hepatocytes

TGF-β is produced by immune cells such as liver macrophages to directly promote fibrosis [24, 25]. In primary KCs and hepatocytes, LPS increased TGF-β secretion, but this effect was attenuated by lobeglitazone (Fig 6A). Therefore, we further investigated the effect of lobeglitazone on TGF-β-induced expression of the inflammasome-specific cytokine IL1β and the fibrosis marker CTGF. Lobeglitazone attenuated TGF-β-induced IL1β mRNA and protein expression and suppressed TGF-β-induced CTGF expression (Fig 6B and 6C). These results indicate that lobeglitazone inhibits LPS-induced inflammasome activation and TGF-β production, thereby suppressing liver fibrosis.

## Discussion

In this study, lobeglitazone decreased LPS-induced NLPR3 inflammasome activation and proinflammatory cytokine production in primary KCs and hepatocytes. Cytokines secreted by activated KCs increased hepatocyte inflammation and HSC activation, and these responses

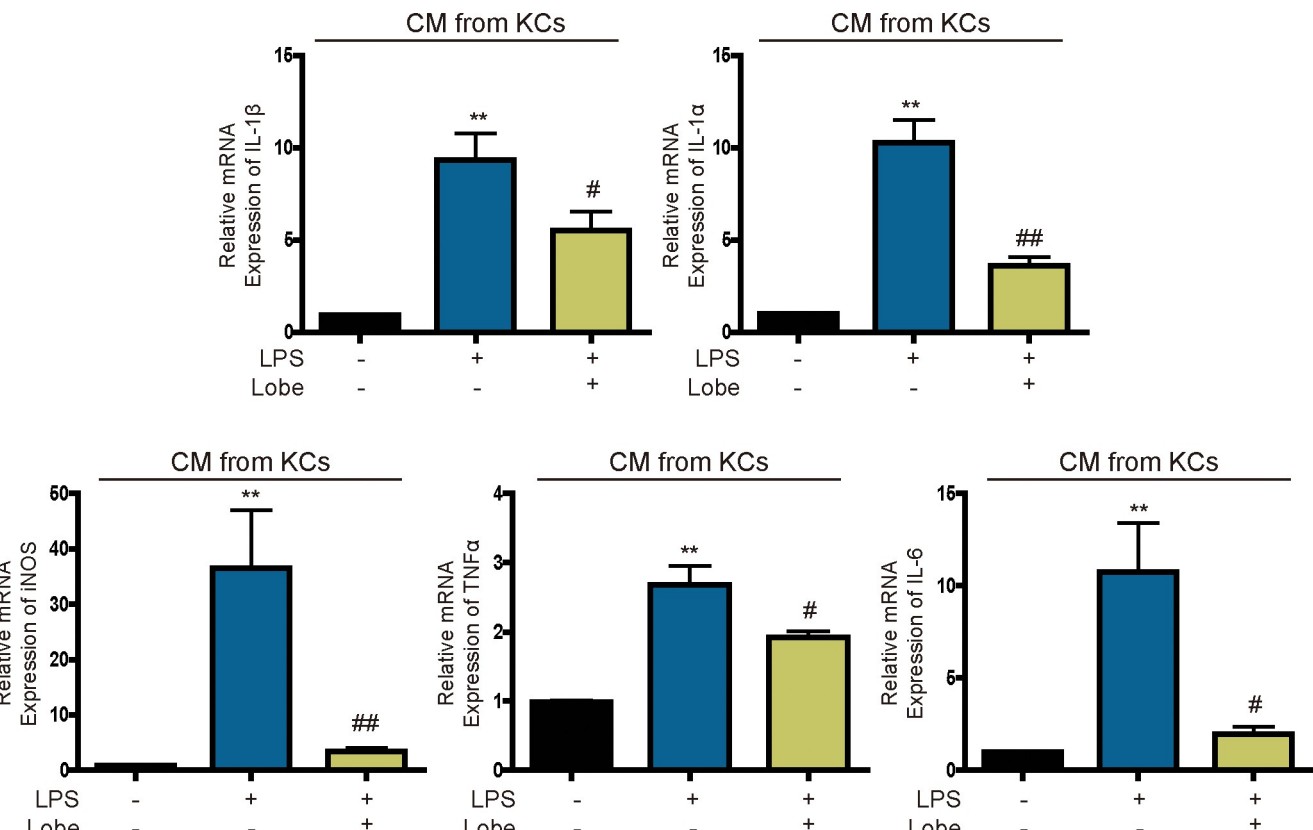

**Fig 3. Lobeglitazone inhibits LPS-induced cytokine secretion by KCs.** Representative real-time RT-PCR analysis of the expression levels of IL1β, IL1α, iNOS, TNFα and IL6 in primary hepatocytes treated with CM of LPS-stimulated KCs. Data are mean ± SEM. **P < 0.01 versus the control; #P < 0.05, ##P < 0.01 versus the LPS-treated group.

were inhibited by lobeglitazone. In addition, lobeglitazone inhibited LPS-induced TGF-β secretion and TGF-β-induced CTGF expression, thereby suppressing liver fibrosis.

KCs are liver-resident macrophages accounting for approximately 10% of all hepatic cells and are involved in the pathogenesis of liver inflammation. The inflammatory response in KCs plays an important role in maintaining liver function and homeostasis in response to various stimuli [6, 26]. The NLRP3 inflammasome is expressed in liver hepatocytes, KCs, and HSCs. These cells are activated under certain conditions to release IL1β [8, 27] Of these cells, the NLRP3 inflammasome is reportedly expressed, assembled, and activated mainly in KCs. Activation of KCs through the NLPR3 inflammasome pathway leads to release of proinflammatory cytokines, hepatocyte injury, and HSC activation [10, 28, 29]. In this study, LPS stimulation of kupffer cells induced significantly higher levels of NLRP3 expression and IL1β secretion, activating the NLRP3 inflammasome, and lobeglitazone inhibited the NLRP3 inflammasome and inflammation. In addition, lobeglitazone suppressed the NLRP3 expression in hepatocytes and HSCs. Lobeglitazone is an agonist of PPARγ, which is a nuclear receptor involved in metabolic regulation [30]. PPARγ has been reported to have anti-inflammatory activity by suppressing proinflammatory cytokines or inducing M2/anti-inflammatory polarization of macrophages [28, 31–34]. In this study, LPS increased expression of the anti-inflammatory cytokines IL10 and Arg1 in KCs, and lobeglitazone attenuated these effects. However, the expression levels of IL10 and Arg1 were significantly lower than those of the proinflammatory cytokines IL1β and iNOS (S1 Fig). This finding is thought to be because KCs have a mixed proinflammatory/anti-

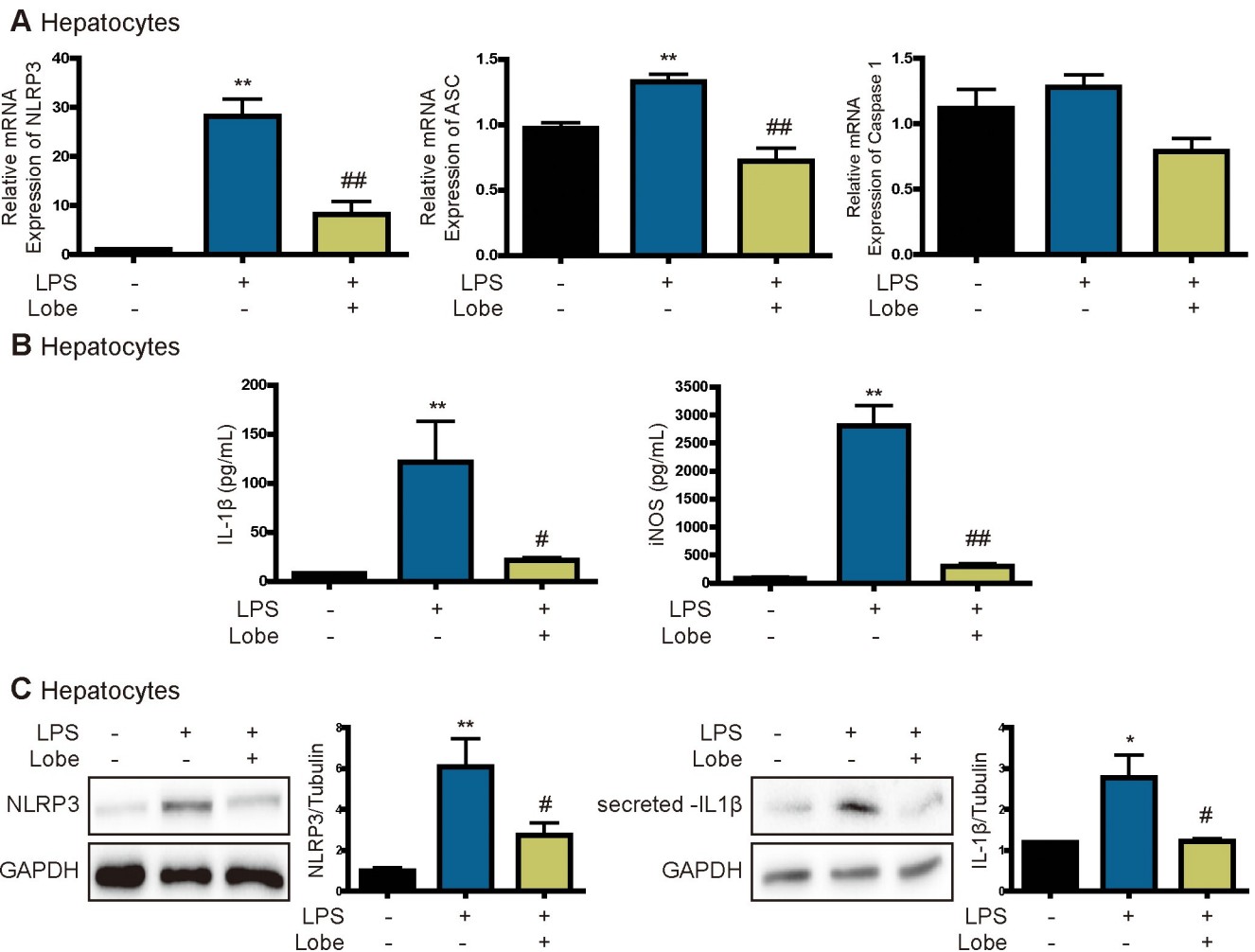

**Fig 4. Lobeglitazone inhibits the LPS-induced NLRP3 inflammasome pathway in hepatocytes.** Primary hepatocytes were pretreated with LPS for 2 h and then treated with lobeglitazone for 24 h. (A) Representative real-time RT-PCR analysis of the expression levels of NLRP3, ASC, and caspase 1. Data are mean ± SEM. **P < 0.01 versus the control, ##P < 0.01 versus the LPS-treated group. (B) After 24 h, the media were collected and analyzed by ELISAs. Data are mean ± SEM of three independent measurements. **P < 0.01 versus the control, #P < 0.05, ##P < 0.01 versus the LPS-treated group. (C) Western blot analysis of the effects of lobeglitazone on LPS-induced NLRP3 and IL1β expression. Data in the bar graphs are mean ± SEM. *P < 0.05, **P < 0.01 versus the control, #P < 0.05 versus the LPS-treated group.

inflammatory phenotype upon LPS treatment, and it is interpreted that M2/anti-inflammatory polarization does not occur upon lobeglitazone treatment. The PPARγ agonists rosiglitazone and pioglitazone were reported to attenuate NLRP3 inflammasome activation in the kidneys and intestines [35–37]. Therefore, we investigated whether the inhibitory effect of lobeglitazone on the NLRP3 inflammasome in the liver is related to PPARγ. Lobeglitazone still reduced NLRP3 inflammasome activity in cells transfected with siPPARγ (S2 Fig). Therefore, the inhibitory effect of lobeglitazone on the NLRP3 inflammasome is independent of PPARγ, at least in primary hepatocytes, and lobeglitazone directly affects the liver inflammasome. However, animal inflammation models are complex and therefore, additional animal studies are needed to determine how lobeglitazone modulates the inflammasome in vivo.

Inflammatory signaling is a complex process regulated by many different components. LPS upregulates the expression of proinflammatory cytokines such as IL1β, IL6, and TNFα, and upregulated IL6 activates STAT3 pathways [38, 39]. LPS-induced proinflammatory cytokine

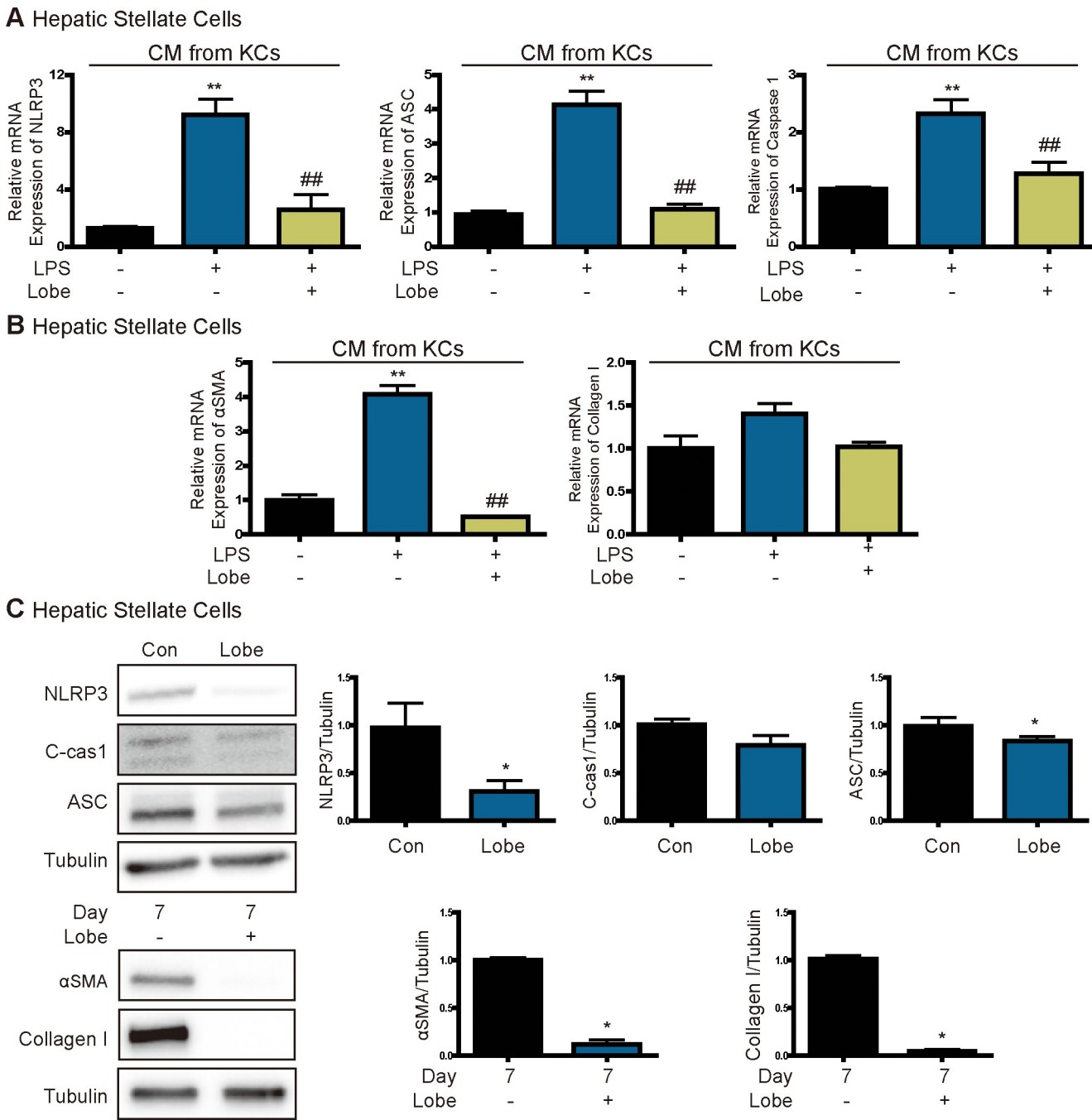

**Fig 5. Lobeglitazone inhibits the NLRP3 inflammasome pathway in HSCs.** (A, B) Representative real-time RT-PCR analysis of the expression levels of NLRP3, ASC, caspase 1, αSMA, and collagen in primary HSCs treated with CM of LPS-stimulated KCs. Data are mean ± SEM. **P < 0.01 versus the control, ##P < 0.01 versus the LPS-treated group. (C) Western blot analysis of inflammasome and fibrosis markers in cultured HSCs. Primary HSCs were cultured for 7 days in DMEM containing 0.5% FBS with or without lobeglitazone. Data in the bar graphs are mean ± SEM. *P < 0.05 versus the control.

expression induced STAT3 phosphorylation in KCs and hepatocytes, but these effects were abrogated by lobeglitazone (S3 Fig). This suggests that STAT3 inhibition by lobeglitazone may participate in its anti-inflammatory effect on the liver.

Cytokines secreted by KCs affect production of proinflammatory cytokines in hepatocytes and activate HSCs to induce ECM secretion, which affects liver fibrosis [3, 40]. In this study,

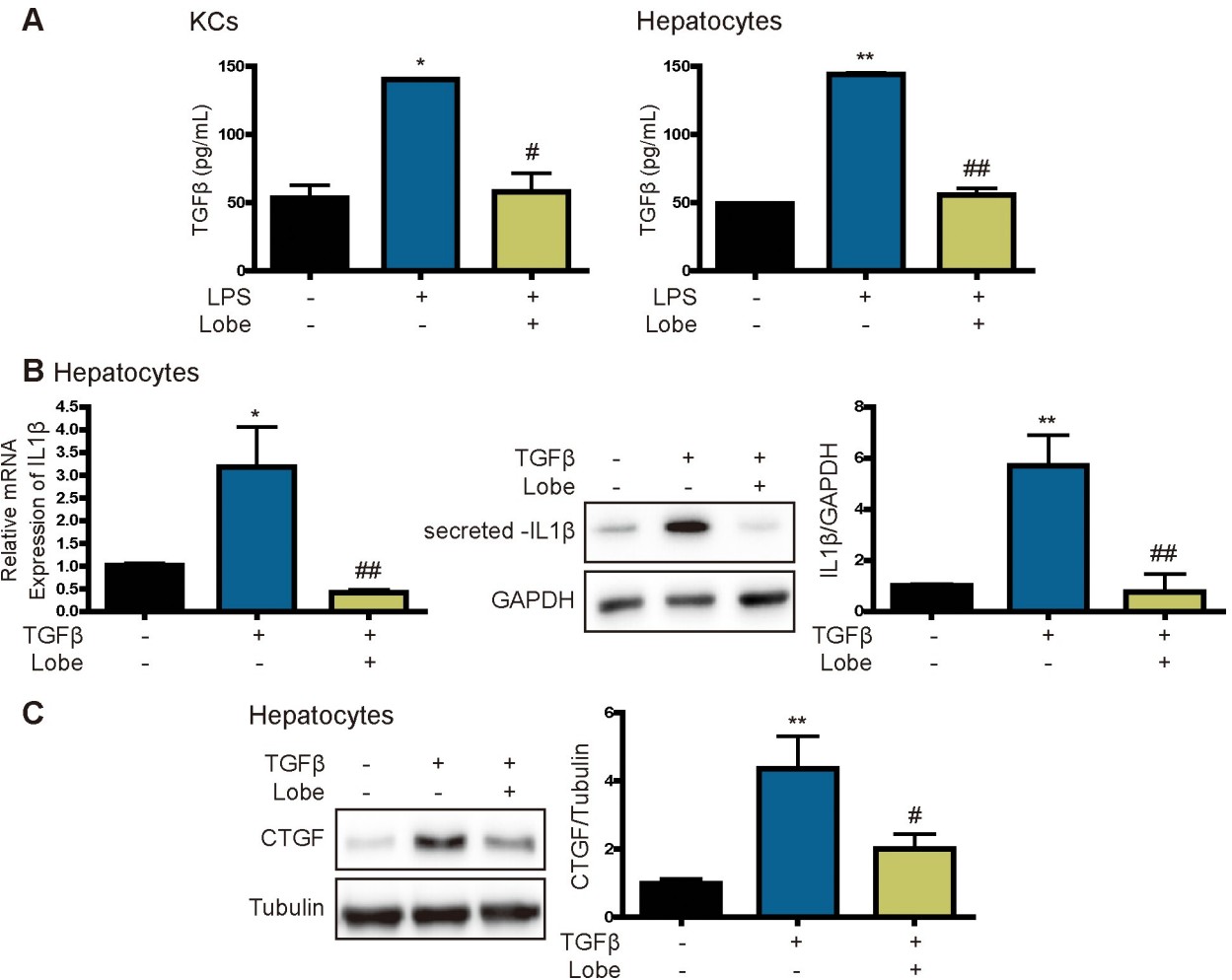

**Fig 6. Lobeglitazone inhibits TGF-β-induced IL1β and CTGF expression.** Primary KCs and hepatocytes were pretreated with LPS for 2 h and then treated with lobeglitazone for 24 h. (A) After 24 h, the media were collected, and TGF-β was measured using an ELISA. Data are mean ± SEM of three independent measurements. *P < 0.05, **P < 0.01 versus the control, #P < 0.05, ##P < 0.01 versus the LPS-treated group. (B) Representative real-time RT-PCR analysis of the expression levels of IL1β in hepatocytes. Data are mean ± SEM. *P < 0.05, **P < 0.01 versus the control, ##P < 0.01 versus the TGF-β-treated group. (C) Western blot analysis of the effects of lobeglitazone on TGF-β-induced CTGF expression. Data in the bar graphs are mean ± SEM. **P < 0.01 versus the control, #P < 0.05 versus the TGF-β-treated group.

several cytokines, such as iNOS and TGF-β, were secreted by LPS-activated KCs, and they increased expression of proinflammatory cytokines in hepatocytes and activated HSCs. However, lobeglitazone suppressed these effects. We examined whether lobeglitazone is directly involved in HSC activation. Lobeglitazone directly inhibits activation of HSCs, but the underlying mechanism requires further study. In addition, lobeglitazone inhibited LPS-induced TGF-β secretion and TGF-β-induced CTGF expression. These results suggest that lobeglitazone has anti-inflammatory and anti-fibrotic effects.

In conclusion, our results show that lobeglitazone decreases LPS-induced NLPR3 inflammasome activation and proinflammatory cytokine production. Additionally, the inhibitory effect of lobeglitazone on inflammasome activation is associated with inhibition of liver fibrosis. These results suggest that lobeglitazone may be a treatment option for inflammation and fibrosis in the liver.

## Supporting information

**S1 Fig. Effects of lobeglitazone on mRNA expression of IL1β, iNOS, IL10 and Arg1 in KCs.** Representative real-time RT-PCR analysis of the expression levels of IL1β, iNOS, IL10, and Arg1.
(JPG)

**S2 Fig. The inhibitory effect of lobeglitazone on the NLRP3 inflammasome is independent of PPARγ.** (A) Real-time RT-PCR analysis of the effect of PPARγ depletion on iNOS and IL1β expression. Primary KCs were transfected with 50 nM siPPARγ or control siRNA (siCon). *$P < 0.05$ versus the control, #$P <$ ##$P < 0.01$ versus the LPS-treated siCon, † $P < 0.05$, †† $P < 0.01$ versus the LPS-treated siPPARγ. (B) Western blot analysis of the effects of PPARγ depletion on NLRP3 expression. ##$P < 0.01$ versus the LPS-treated siCon, † $P < 0.05$ versus the LPS-treated siPPARγ.
(JPG)

**S3 Fig. Lobeglitazone inhibits LPS-induced p-STAT3 expression.** (A, B) Primary KCs and hepatocytes were pretreated with LPS for 2 h and then treated with lobeglitazone for 24 h. Western blot analysis of the effects of lobeglitazone on LPS-induced p-STAT3 expression. Data in the bar graphs are mean ± SEM. **$P < 0.01$ versus the control, ##$P < 0.01$ versus the LPS-treated group.
(JPG)

**S1 Raw images.**
(PDF)

## Author Contributions

**Conceptualization:** Hye-Young Seo, Mi Kyung Kim, Byoung Kuk Jang.

**Data curation:** Hye-Young Seo, So-Hee Lee, Ji Yeon Park, Sol Han.

**Formal analysis:** Eugene Han.

**Funding acquisition:** Hye-Young Seo, Mi Kyung Kim, Byoung Kuk Jang.

**Investigation:** Eugene Han, Jae Seok Hwang.

**Methodology:** So-Hee Lee.

**Supervision:** Mi Kyung Kim, Byoung Kuk Jang.

**Validation:** So-Hee Lee.

**Visualization:** So-Hee Lee.

**Writing – original draft:** Hye-Young Seo.

**Writing – review & editing:** Mi Kyung Kim, Byoung Kuk Jang.

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
