## [Decision Letter · Decision Letter 0]

18 Jun 2023

PONE-D-23-14508Lobeglitazone inhibits LPS-induced NLRP3 inflammasome activation and inflammation in the liverPLOS ONE

Dear Dr. Jang,

Thank you for submitting your manuscript to PLOS ONE. After careful consideration, we feel that it has merit but does not fully meet PLOS ONE’s publication criteria as it currently stands. Therefore, we invite you to submit a revised version of the manuscript that addresses the points raised during the review process.

The article is about the roles of Lobeglitazone on LPS-induced NLRP3 activation in Kupffer cells, hepatocytes, and Hepatic stellate cells. The evaluations among the reviewers are quite different but I think the manuscript can be reevaluated after major revisions. One reviewer gave some critical questions and please respond to them carefully.

We look forward to receiving your revised manuscript.

Kind regards,

Kenji Fujiwara, PhD, MD

Academic Editor

PLOS ONE

Journal Requirements:

"This research was supported by the Basic Science Research Program through the National Research Foundation of Korea (NRF) funded by the Ministry of Education (NRF‐2021R1I1A3046593, NRF‐ 2021R1I1A3059150), and was supported by a NRF grant funded by the Korea government (MIST) (NRF‐2022R1A2C1006416). (BK Jang, MK Kim, HY Seo)"

Additional Editor Comments:

Dear Dr. Jang.

The article is about the roles of Lobeglitazone on LPS-induced NLRP3 activation in Kupffer cells, hepatocytes, and Hepatic stellate cells. The evaluations among the reviewers are quite different but I think the manuscript can be reevaluated after major revisions. One reviewer gave some critical questions and please respond to them carefully.

I am looking forward to resubmission.

Best regards,

Kenji Fujiwara

Reviewers' comments:

Reviewer's Responses to Questions

**Comments to the Author**

1. Is the manuscript technically sound, and do the data support the conclusions?

Reviewer #1: Partly

Reviewer #2: Partly

Reviewer #3: Yes

2. Has the statistical analysis been performed appropriately and rigorously? 

Reviewer #1: Yes

Reviewer #2: Yes

Reviewer #3: Yes

3. Have the authors made all data underlying the findings in their manuscript fully available?

Reviewer #1: Yes

Reviewer #2: No

Reviewer #3: Yes

4. Is the manuscript presented in an intelligible fashion and written in standard English?

Reviewer #1: Yes

Reviewer #2: Yes

Reviewer #3: Yes

5. Review Comments to the Author

Reviewer #1: This study investigated the roles of Lobeglitazone, a peroxisome proliferator-activated receptor gamma (PPARgamma) ligand, on LPS-induced NLRP3 activation in primary Kupffer cells (KCs) and hepatocytes and fibrosis on stellate cells. The authors need to consider the following issues.

1. PPARgamma has been reported to be responsible for the M2/anti-inflammatory polarization of macrophages (PMID: 33627030). I suggest the authors investigate the status of KCs polarization.

2. The authors investigated the roles of Lobeglitazone on LPS-induced NLRP3 inflammasome activation in primary hepatocytes. However, NLRP3 inflammasome has been suggested to be mainly assembled in macrophages, and hepatocytes have been demonstrated to be resistant to pyroptosis due to the low expression of caspase 1/11.

3. Figure 5- the authors need to investigate the roles of Lobeglitazone on CM-treated HSCs.

4. Figure 6-the authors found that Lobeglitazone could downregulate LPS-induced up-regulation of TGF-β. Then they focused on the roles of TGF-β on IL1β and CTGF on hepatocytes. However, if they want to reveal the roles of Lobeglitazone on the activation of HSCs, they need to investigate the effects of Lobeglitazone on HSCs but not hepatocytes.

Reviewer #2: Seo et al., found lobegilitazone inhibit LPS induced NLRP3 inflammasome activation and inflammatin in the liver. But the resluts was worried. Activation of NLRP3 inflammasomes at the cellular level requires reagents, otherwise they cannot be activated, but this paper does not provide them. There are no positive drugs, and the dosage of the drugs should be at least two.

Reviewer #3: The manuscript was technically sound, the conclusions were supported by data, and appropriate statistical methods were used. The article is written in standard English, the language is clear, concise and to the point. Accept the article。

6. PLOS authors have the option to publish the peer review history of their article (what does this mean?). If published, this will include your full peer review and any attached files.

Reviewer #1: **Yes: **TAO ZENG

Reviewer #2: No

Reviewer #3: No

---

## [Author Response · Author response to Decision Letter 0]

26 Jul 2023

Reviewer #1: This study investigated the roles of Lobeglitazone, a peroxisome proliferator-activated receptor gamma (PPARgamma) ligand, on LPS-induced NLRP3 activation in primary Kupffer cells (KCs) and hepatocytes and fibrosis on stellate cells. The authors need to consider the following issues.

Thank you for your comments to improve this manuscript.

1. PPARgamma has been reported to be responsible for the M2/anti-inflammatory polarization of macrophages (PMID: 33627030). I suggest the authors investigate the status of KCs polarization.

- Thank you for your good suggestion. It has been reported that (PMID: 33627030) (Wang, Zhang et al. 2021) PPARgamma is responsible for the M2 polarization of macrophages. 

In the current study, we also found that the anti-inflammatory cytokine IL-10 and Arg1 gene expression was increased in kupffer cells with LPS treatment and decreased with lobeglitazone treatment. However, the expression level of IL10 and Arg1 were significantly lower than that of the pro-inflammatory cytokines IL1beta and iNOS. 

The opposite result is thought to be because kupffer cells have a mixed proinflammatory/anti-inflammatory phenotype under LPS treatment, and it is interpreted that M2/anti-inflammatory polarization did not appear during lobeglitazone treatment. Thus, modulation of lobeglitazone could reduce LPS-nduced M1 predominant kupffer cells polarization and thereby reduce the liver inflammatory response.

We have modified the discussion and supplementary figure (S1 Fig.) (Line 211)

- In this reference paper (Wang, Zhang et al. 2021), they report that Nrf2 can regulate macrophage polarization. Specifically, activation of Nrf2 can block pro-inflammatory cytokine and chemokine production induced by M1 stumulation and shift polarization of macrophages to M2. 

In our results, lobeglitazone increased the expression of HO-1 in primary hepatocytes. However, although lobeglitazone did not increase HO-1 by siNRF2 transfection, iNOS still decreased, suggesting that the inhibitory effect of lobeglitazone on inflammation is not related to the NRF2/HO1 pathway.

2. The authors investigated the roles of Lobeglitazone on LPS-induced NLRP3 inflammasome activation in primary hepatocytes. However, NLRP3 inflammasome has been suggested to be mainly assembled in macrophages, and hepatocytes have been demonstrated to be resistant to pyroptosis due to the low expression of caspase 1/11.

- As the reviewer suggested, Kupffer cells are the major producers of NLRP3 and IL-1β after LPS stimulation. In our results show that Lobeglitazone inhibits LPS-induced expression of pro-inflammatory cytokines and NLRP3 inflammasome in KC. 

- Hepatocytes are resistant to pyroptosis due to low expression of caspase 1/11 (Sun, Zhong et al. 2022). Therefore, as pointed out by the reviewer, it is difficult to explain the association with pyroptosis. However, in this study, LPS-induced NLRP3 and IL1beta secretion was increased in primary hepatocytes, and their expression was decreased by lobeglitazone. 

3. Figure 5- the authors need to investigate the roles of Lobeglitazone on CM-treated HSCs.

As the reviewer suggested, KC activated by LPS secretes several cytokines, such as iNOS and TGFβ, and the several cytokine of KCs activates HSCs (Slevin, Baiocchi et al. 2020) (Tacke and Zimmermann 2014) (Sun, Xiu et al. 2018). In our results, LPS increased the secretion of iNOS (Figure 1) and TGFβ (Figure 6A) in KC, and the conditioned media containing these cytokines increased HSC activation. Lobeglitazone may also reduce HSC activation by reducing iNOS and TGFβ secretion.

We have modified the discussion (Line 233)

4. Figure 6-the authors found that Lobeglitazone could downregulate LPS-induced up-regulation of TGF-β. Then they focused on the roles of TGF-β on IL1β and CTGF on hepatocytes. However, if they want to reveal the roles of Lobeglitazone on the activation of HSCs, they need to investigate the effects of Lobeglitazone on HSCs but not hepatocytes.

Thank you for your good suggestion. Cytokines secreted by hepatocyte damage and activated KC activate HSCs. Therefore, it is thought that inhibiting the inflammatory process in hepatocytes and KCs can inhibit HSC activation. In addition, lobeglitazone has the effect of directly inhibiting HSC activation. However, the mechanism needs further study.

Reviewer #2: Seo et al., found lobegilitazone inhibit LPS induced NLRP3 inflammasome activation and inflammatin in the liver. But the resluts was worried. Activation of NLRP3 inflammasomes at the cellular level requires reagents, otherwise they cannot be activated, but this paper does not provide them. There are no positive drugs, and the dosage of the drugs should be at least two.

Thanks for the reviewer's comments. The inflammasome condition is either LPS+ATP or LPS+nigericin. In primary KC, LPS and LPS+nigericin treatment were compared, but LPS alone or LPS+nigericin increase NLRP3 and IL1beta, and there was no significant difference between the two groups.

In addition, a paper reported that NLRP3 inflammasome is increased by LPS treatment (Wang, 

Wang et al. 2021) (Fan, Li et al. 2021) (Fan, Li et al. 2022) (Yu, Lan et al. 2017). In this study, LPS treatment also increased inflammasomes by increasing NLRP3, secreted IL1β, and cleaved caspase 1 in KC (Figure 2B).

Reviewer #3: The manuscript was technically sound, the conclusions were supported by data, and appropriate statistical methods were used. The article is written in standard English, the language is clear, concise and to the point. Accept the article。

Thank you for reviewing our paper.

References

Fan, G., Y. Li, J. Chen, Y. Zong and X. Yang (2021). "DHA/AA alleviates LPS-induced Kupffer cells pyroptosis via GPR120 interaction with NLRP3 to inhibit inflammasome complexes assembly." Cell Death & Disease 12(1): 73.

Fan, G., Y. Li, Y. Liu, X. Suo, Y. Jia and X. Yang (2022). "Gondoic acid alleviates LPS‑induced Kupffer cells inflammation by inhibiting ROS production and PKCθ/ERK/STAT3 signaling pathway." International Immunopharmacology 111: 109171.

Slevin, E., L. Baiocchi, N. Wu, B. Ekser, K. Sato, E. Lin, L. Ceci, L. Chen, S. R. Lorenzo, W. Xu, K. Kyritsi, V. Meadows, T. Zhou, D. Kundu, Y. Han, L. Kennedy, S. Glaser, H. Francis, G. Alpini and F. Meng (2020). "Kupffer Cells: Inflammation Pathways and Cell-Cell Interactions in Alcohol-Associated Liver Disease." The American Journal of Pathology 190(11): 2185-2193.

Sun, L., M. Xiu, S. Wang, D. R. Brigstock, H. Li, L. Qu and R. Gao (2018). "Lipopolysaccharide enhances TGF-β1 signalling pathway and rat pancreatic fibrosis." Journal of Cellular and Molecular Medicine 22(4): 2346-2356.

Sun, P., J. Zhong, H. Liao, P. Loughran, J. Mulla, G. Fu, D. Tang, J. Fan, T. R. Billiar, W. Gao and M. J. Scott (2022). "Hepatocytes Are Resistant to Cell Death From Canonical and Non-Canonical Inflammasome-Activated Pyroptosis." Cellular and Molecular Gastroenterology and Hepatology 13(3): 739-757.

Tacke, F. and H. W. Zimmermann (2014). "Macrophage heterogeneity in liver injury and fibrosis." Journal of Hepatology 60(5): 1090-1096.

Wang, X., L. Wang, R. Dong, K. Huang, C. Wang, J. Gu, H. Luo, K. Liu, J. Wu, H. Sun and Q. Meng (2021). "Luteolin ameliorates LPS-induced acute liver injury by inhibiting TXNIP-NLRP3 inflammasome in mice." Phytomedicine 87: 153586.

Wang, Y. R., X. N. Zhang, F. G. Meng and T. Zeng (2021). "Targeting macrophage polarization by Nrf2 agonists for treating various xenobiotics-induced toxic responses." Toxicol Mech Methods 31(5): 334-342.

Yu, X., P. Lan, X. Hou, Q. Han, N. Lu, T. Li, C. Jiao, J. Zhang, C. Zhang and Z. Tian (2017). "HBV inhibits LPS-induced NLRP3 inflammasome activation and IL-1β production via suppressing the NF-κB pathway and ROS production." J Hepatol 66(4): 693-702.

---

## [Decision Letter · Decision Letter 1]

11 Aug 2023

Lobeglitazone inhibits LPS-induced NLRP3 inflammasome activation and inflammation in the liver

PONE-D-23-14508R1

Dear Dr. Jang,

We’re pleased to inform you that your manuscript has been judged scientifically suitable for publication and will be formally accepted for publication once it meets all outstanding technical requirements.

Kind regards,

Kenji Fujiwara, PhD, MD

Academic Editor

PLOS ONE

Additional Editor Comments (optional):

Dear Dr. Jang.

Thank you for revising your manuscript appropriately. The reviewer and I agreed to the acceptance.

Yours sincerely,

Kenji Fujiwara

Academic editor

Reviewers' comments:

Reviewer's Responses to Questions

**Comments to the Author**

1. If the authors have adequately addressed your comments raised in a previous round of review and you feel that this manuscript is now acceptable for publication, you may indicate that here to bypass the “Comments to the Author” section, enter your conflict of interest statement in the “Confidential to Editor” section, and submit your "Accept" recommendation.

Reviewer #1: All comments have been addressed

2. Is the manuscript technically sound, and do the data support the conclusions?

Reviewer #1: Yes

3. Has the statistical analysis been performed appropriately and rigorously? 

Reviewer #1: Yes

4. Have the authors made all data underlying the findings in their manuscript fully available?

Reviewer #1: Yes

5. Is the manuscript presented in an intelligible fashion and written in standard English?

Reviewer #1: Yes

6. Review Comments to the Author

Reviewer #1: (No Response)

7. PLOS authors have the option to publish the peer review history of their article (what does this mean?). If published, this will include your full peer review and any attached files.

Reviewer #1: No

---

## [Editor Report · Acceptance letter]

16 Aug 2023

PONE-D-23-14508R1 

Lobeglitazone inhibits LPS-induced NLRP3 inflammasome activation and inflammation in the liver 

Dear Dr. Jang:

I'm pleased to inform you that your manuscript has been deemed suitable for publication in PLOS ONE. Congratulations! Your manuscript is now with our production department. 

Kind regards, 

on behalf of

Dr. Kenji Fujiwara 

Academic Editor

PLOS ONE